# Technological Innovation, Risk-Taking and Firm Performance—Empirical Evidence from Chinese Listed Companies

**Hui Zhang *** and **Vesarach Aumeboonsuke**

National Institute of Development Administration, Bangkok 10240, Thailand
* Correspondence: zhimcf98@126.com

**Abstract:** Technological innovation can restructure the production factors of enterprises, and it is an important factor for enterprises to meet market demand, improve competitiveness, form long-term competitive advantages and obtain sustainable development. This study focuses on the practical issue of the impact of technology innovation on firm performance. Taking 1166 listed companies in China from 2012 to 2020 as research samples, this study systematically investigates and reveals the impact of technological innovation on firm performance and its internal impact mechanism. The research shows that technological innovation significantly reduces firm performance, and that conclusion holds after an endogeneity test and a robustness test. The analysis of the impact mechanism shows that risk-taking is an important transmission path of corporate technological innovation affecting corporate performance and that technological innovation reduces firm performance by improving the risk-taking capacity. Finally, a heterogeneity test regarding the firm ownership shows that technological innovation has a significantly stronger negative impact on the performance of non-state-owned enterprises than on that of state-owned enterprises. The relevant government departments and market subjects should fully understand and give attention to the impact of enterprise technological innovation on firm performance and its mechanism, which has important practical significance for standardizing and strengthening enterprise R&D management, reducing the market and technological risks of firm technological innovation and perfecting modern enterprise systems. It is helpful for firms to form a sustainable technology innovation cycle development mode.

**Keywords:** technological innovation; risk-taking; firm performance; firm ownership

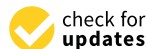



## 1. Introduction

Nowadays, the competition in business is fierce. Whether it is the implementation of the national innovation-driven strategy or the transformation of traditional industries into the new pattern of "dual circulation" development, firms are forced to embark on the pursuit of technological innovation to win the core competitiveness of sustainable development [1]. Enterprise technology innovation has become an important driving force for China's high-quality economic development. However, some scholars have found that the technological innovation of Chinese enterprises is characterized by high quantity, low quality and policy catering and that it lacks substantive innovation, bringing little benefit to the development of enterprises [2]. As investment in technological innovation increases, the uncertainty, risks and challenges faced by enterprises also increase. Due to the influence of scale, capital, systems and other factors, private enterprises are more likely to fall into a state of excessive risk-taking than state-owned enterprises, and their inadequate ability to cope with risk-taking leads to a further reduction in firm performance. Risk and performance are two sides of the same coin. The effectiveness of a firm's technological innovation should be evaluated not only by its ability to improve the corporate performance but also by the risky consequences that accompany the pursuit of performance [3]. What role does technological



innovation play in the relationship between firm performance and risk-taking? Does firm performance increase with risk, does it increase with a decrease in risk or does it decrease with an increase in risk? Enterprises need to pay attention to the influence mechanism of technological innovation on enterprise risk-taking and firm performance.

Technological innovation is an important way to differentiate enterprise products and services, and it is key to the improvement of the enterprise economic performance [4]. Technological innovation refers to creative activities carried out by enterprises to acquire high-level, new technologies and R&D investments made from human resources, property and other factors to achieve core competitiveness in the corresponding field [5]. Research and development (R&D), as necessary expenditures of enterprises, are mainly used to guarantee a source of funds for enterprise innovation. Therefore, most studies regard R&D input as a measurement index of enterprise technological innovation [6]. Since Schumpeter put forward innovation theory, the relationship between enterprise innovation and firm performance has been widely studied. Due to differences in enterprise culture and capital structures, the findings of these studies do not reflect a clear conclusion. Most scholars believe that there is a significant positive correlation between technological innovation and firm performance. The more innovation an enterprise engages in, the more profit it will make [7–9]. After this, in-depth research was conducted in this subfield, for example, grouping samples and comparing the relationship between innovation and performance across different groups. State-owned enterprises are supported by government systems and policies and have more resources but lack an awareness of technological innovation. However, as the main force in responding to policies of mass entrepreneurship and innovation, private enterprises are stronger than state-owned enterprises in terms of their return on R&D investments and technological innovation, but they also face a higher level of risk [10,11]. In the context of corporate governance, firms' technological innovation is affected by its equity structure, incentive mechanism, internal and external governance environment, labor capital input and other factors, so innovation input and output also significantly differ [12,13].

Some scholars have come to the opposite conclusion in their studies on the relationship between enterprise technological innovation and firm performance. Innovation activities bring great uncertainty, and enterprises are susceptible to interference from external environmental fluctuations, which have a negative impact on their economic development [14]. There is not a simple linear relationship between enterprise technological innovation and firm performance, and there are differences across different market environments and different subjects, because both opportunities and risks are brought by innovation. Through a literature review, Gerben found that only 20% of project innovations are feasible and effective [15], and 39% of innovation projects end in failure [16]. This means that one-third of enterprises' investments in innovation do not bring better development opportunities but entail a large amount of capital and resource loss, resulting in a reduction in firm performance [17]. Innovation is a complex issue that is often affected by the environment where the subject of innovation is located. Regions that are highly suitable for technological innovation contribute more to economic growth, and technological innovation needs time to form and requires support from economic development; indeed, every region and enterprise should not blindly increase its innovation input [18].

Risk-taking refers to the ability of an enterprise to withstand risks in the face of uncertainties in investment or business activities, and it is most often measured based on factors such as the following: earnings, stock volatility, R&D, capital expenditures and debt ratios [19–21]. Many studies on enterprise management and entrepreneurship have discussed the relationship between technological innovation and risk-taking [22]. Enterprise innovation activities are characterized by high costs, high risk and long benefit cycles. Enterprise managers consider the problem of risk aversion when making project decisions to maximize the profit generated by innovation input [23]. To quickly increase the value of enterprises, managers tend to take higher risks and pursue innovation to achieve long-term development [20]. However, when engaging in innovation and development,

enterprises should carry out risk assessments carefully. Neither risk aversion nor excessive risk chasing is advisable. Risk balance is the best and most difficult approach for enterprises. In their studies on the technological innovation and risk-taking of enterprises, many scholars have used data from commercial banks to conduct research and confirmed that the innovation and development of science and technology in the context of financial services reducing the risk-taking of commercial banks. This effect is more significant in larger-scale banks, those with better foundations or those with state ownership [24,25]. Sonia et al. took typical innovative companies in Sweden as their research object and found that both innovation-supporting activities and innovation goals have positive impacts on risk-taking, while the innovation process has a significant negative impact on enterprise risk-taking and innovation performance [26]. Therefore, innovation is a complex process of accumulation. Fluctuations in market conditions can vary between firms and sometimes even have diametrically opposite effects.

In summary, the literature has made some achievements in the study of the relationships between technological innovation, firm performance and risk-taking. However, due to different factors such as capital structure and industry, the conclusions drawn are not the same [7–9,26]. Scholars have mostly studied the relationship between risk-taking and the mediating role of risk-taking from the perspectives of executive characteristics, equity governance and executive compensation [27–29]. Few studies have taken risk-taking as a mediator to study the role of risk-taking in the relationship between technological innovation and firm performance. Results based on different research perspectives, research objectives and research subjects are often not corroborated or even contradictory [12,13]. Thus, this study selected 1166 listed companies in China from 2012 to 2020 as research samples to explore the relationship between technological innovation, risk-taking and corporate performance. Thus, based on the existing research, this paper further discusses the influence mechanism of technological innovation on firm performance under the difference of property right nature, and this study selects state-owned enterprises and non-state-owned enterprises as research objects and conducts a dialectical analysis of the impact of technological innovation in different organizational situations to provide practical guidance for the technological innovation behavior of state-owned and non-state-owned enterprises in uncertain environments. In recent years, China has been in an important period of transformation and upgrading with rapid economic growth. Discussing the impact of technological innovation on Chinese enterprise performance, it is conducive to the long-term and stable improvement of Chinese enterprise performance, the maintenance of sustainable economic development of enterprises and the promotion of the international competitiveness of enterprises. At the same time, the research from the new perspective of risk-taking enriches the existing theories and provides a theoretical basis for enterprises in other developing countries to carry out technological innovation and change reasonably and effectively.

The research contribution of this paper is mainly reflected in the following aspects. First, this study clearly reveals the mechanism of the impact of technological innovation on firm performance and further discusses this impact in the context of the firm ownership difference between state-owned enterprises and non-state-owned enterprises. Second, this study includes technological innovation, risk-taking and firm performance with a logical framework and functional path. While managers focus on the relationship between technological innovation and enterprise performance, the relationships between the three aforementioned factors are analyzed from the new perspective of risk-taking. This will help enterprises reasonably choose between risk avoidance and risk pursuit, broaden their research horizons and provide theoretical guidance for the implementation of technological innovation activities in Chinese enterprises under the "new normal" of economic development.

## 2. Theoretical Analysis and Research Hypothesis

Schumpeter was the first economist to put forward a systematic and complete theory of innovation for Western economics. Subsequent theories of innovation and institutional

innovation have been developed upon, and extended from, this basis. According to Schumpeter's theory, the fundamental purpose of innovation is to obtain potential social benefits. When entrepreneurs realize this, they actively invest or attract capital and create or introduce new production modes to generate profits. Therefore, sufficient R&D investment is a prerequisite for enterprises to carry out innovation activities, and the generation of monopoly profits and innovation profits is the primary reason why enterprises promote innovation. For enterprises, R&D investment is only the means by which the goals of high output and high performance are reached. Does enterprise technology innovation truly improve performance? In fact, due to variations in the level of R&D and management across different enterprises, it is unclear whether enterprises can successfully convert innovation input into enterprise income [30].

In contrast to general investment activities, the technological innovation of enterprises is characterized by extensive periodicity, uncertainty and high risk. First, in the initial stage of innovation investment, the marginal productivity of R&D investment is low, and it is difficult for enterprises to generate substantial output returns [31]. It takes a long time to transform investment in innovation to the commercialization of innovation results. Moreover, innovation diffusion theory holds that, even if an enterprise makes a technological breakthrough, it takes time to gradually expand the market scale of new products [32,33]. When enterprises do not generate enough profits from innovation activities, it is difficult for them to offset the associated high upfront costs, which have a negative impact on firm performance. Second, an enterprise's innovation activities represent its exploration of the feasibility of new technologies. The more strongly it leads in terms of innovation, the more obvious its innovation characteristics and the more difficult it is to imitate. However, the farther it is from the market demand, the greater the associated risk and the more uncertain the associated benefits. To ensure innovation profits and reduce the benefits of other enterprises that have not engaged in innovation activities related to their own innovation output, enterprises should maintain a certain technological frontier in their innovation activities and increase the difficulty of imitation [34]. Therefore, the innovation input of enterprises must exceed a certain critical value; however, the opportunity cost effect will lead to massive innovation input crowding out the resources of enterprises in other aspects, even reducing their profits [35]. Finally, enterprises exist in an era characterized by exponential technological progress, and managers commonly underestimate the rate of technological obsolescence. Rapid declines in the profit rates of products associated with a particular technology due to the limitations of that technology may exert pressure on managers, who hope to promote the production of next-generation products through innovation activities. Moreover, because it is difficult to judge when to change or discontinue an investment strategy, managers are more inclined to increase their investment in innovation and R&D related to existing technology than to invest in new and unfamiliar technology [36]. However, such technological innovation projects bring high costs and low performance [35]. Thus, it can be concluded that technological innovation may not have a significant promoting effect on firm performance but have a negative effect; that is, technological innovation reduces firm performance [37,38]. Accordingly, hypothesis 1 is proposed:

**Hypothesis 1.** *Technological innovation is negatively correlated with firm performance. Technological innovation inhibits the improvement of firm performance.*

Managers allocate investment funds from tangible assets (such as capital expenditure) to intangible assets (such as research and development), increasing enterprise risks. In contrast to capital expenditures on property, plant and equipment, enterprise expenditures on technological innovation are usually regarded as high-risk investments [39,40]. The literature has shown that R&D investment has a positive effect on risk [41] that was larger than the capital investment's effect on risk [42]. Thus, it can be concluded that technological innovation has a positive risk effect, which means that implementing innovation activities

increases enterprise risk-taking. Enterprises that are willing to take on risk actively engage in actions related to pursuing opportunities and contribute to firm performance by adopting efficient and bold strategies [43]. However, the ability of enterprises to bear risks is limited. Excessive risk-taking without risk control may lead enterprises to be impulsive and aggressive, and problems such as weak opportunity ability and weak operation and management ability may arise. Low-resource utilization efficiency not only is disadvantageous for enterprises in terms of identifying opportunities and making decisions based on market development trends but also affects the size of the market that companies can explore. Similarly, such enterprises are unable to make use of limited resources in business operations and strategy implementation, which is not conducive to the overall internal operation and management of organizations; moreover, they are unable to quickly respond to changes in the external environment and thus experience reduced firm performance [44]. Therefore, a continuous increase in the intensity of technological innovation may cause enterprises to fall into a state of excessive risk-taking, which leads to a reduction in firm performance. Accordingly, hypothesis 2 is proposed:

**Hypothesis 2.** *A firm's risk-taking capacity plays a mediating role in the relationship between technological innovation and firm performance. Technological innovation reduces firm performance by improving a firm's risk-taking capacity.*

In addition, from a dynamic perspective, due to differences in firms' ownership structure, scale and capacity, the innovation activities and levels of different enterprises differ to some extent. Moreover, with the gradual establishment of the socialist market economy system and the improvement of the capital market, the ownership structure of Chinese enterprises has undergone tremendous change, and there has been a trend of mutual penetration and integration of firm ownership. Therefore, the hypothesis of enterprise heterogeneity is the basis for the study of enterprise behavior, and the research of firm innovation behavior without regard to differences in the nature of firm ownership lacks validity and relevance [45]. Based on the special national conditions of China, this study introduces firm ownership to the relationship between technological innovation and firm performance, divides its sample into subsamples of state-owned enterprises and non-state-owned enterprises, further explores the heterogeneous impact of technological innovation on firm performance and investigates whether there are significant differences across different groups. This is conducive to assessing the actual impact of firm ownership systems on enterprise innovation activities, enriching the research content and making it more relevant.

## 3. Research Design

### 3.1. Sample Selection and Data Sources

Relevant data of the listed companies from 2012 to 2020 were selected and processed as follows. (1) According to the 2021 industry classification of the China Securities Regulatory Commission (CSRC), financial companies such as those providing capital market services, monetary and financial services or insurance, as well as firms in other financial industries, were excluded. (2) Companies delisted between 1 January 2012 and 31 December 2020 were excluded, and companies listed after 31 December 2012 were excluded. (3) ST companies, *ST companies and other companies with abnormal financial conditions were excluded. (4) Companies with missing data pertaining to our research variables, such as our explained variables, explanatory variables, mediating variables and control variables, for the examined company/year observations were deleted. After screening, 10,494 observed data of 1166 listed companies were finally obtained as research samples, and stata15.1 was used for the correlation analysis. Among them, the 1166 listed companies included Midea Group, Kweichow Moutai, China Telecom, China Petrochemical and other well-known companies and involved consumer goods, information technology, telecommunications services, industrial, energy, raw materials and manufacturing industries. The sample almost covered

all of China's industry and was representative. The data used in this paper were taken from the CSMAR Database and the China Statistical Yearbook, and the missing data of some indicators were supplemented based on the linear interpolation method.

*3.2. Specification of the Model*

To test the influence of technological innovation capability on firm performance, the following model is constructed:

$$JROA_{i,t} = \alpha_0 + \alpha_1 Innovation_{i,t} + \delta_1 X_{i,t} + \mu_{i1} + \omega_{t1} + \varepsilon_{it1} \tag{1}$$

$$Risk_{i,t} = \beta_0 + \beta_1 Innovation_{i,t} + \delta_2 X_{i,t} + \mu_{i2} + \omega_{t2} + \varepsilon_{it2} \tag{2}$$

$$JROA_{i,t} = \gamma_0 + \gamma_1 Innovation_{i,t} + \gamma_2 Risk_{i,t} + \delta_3 X_{i,t} + \mu_{i3} + \omega_{t3} + \varepsilon_{it3} \tag{3}$$

where $i$ represents the focal firm, $t$ represents the year, $\mu$ represents individual fixed effects, $\omega$ represents time fixed effects, *JROA* represents firm performance, *Innovation* represents the firm's technological innovation ability, *Risk* represents the firm's risk-taking ability, $X$ represents the utilized control variables and $\varepsilon_{i,t}$ is the error term. The mechanism by which technological innovation affects firm performance is shown in Figure 1.

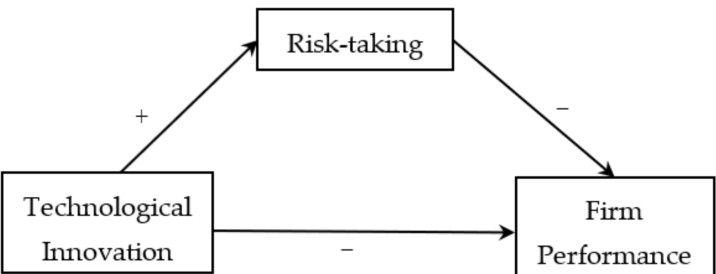

**Figure 1.** Theoretical model of the impact of technological innovation on firm performance.

*3.3. Variable Definition*

3.3.1. Explained Variable

Firm performance (JROA). Most of the existing relevant research literature measures firm performance indicators from the perspectives of market performance (such as Tobin's Q) or financial performance (such as return on sales, return on equity, return on shareholders' equity or return on assets) [46]. Considering that the net interest rate on the total assets index can effectively reflect the profitability of enterprises [47], this index is used to measure the performances of enterprises [48].

3.3.2. Explanatory Variable

Indicators used to measure the technological innovation of enterprises are mainly developed from the perspectives of input and output. Output mainly consists of the number of patents, the sales revenue of new products and the citation rate of patents, while input comprises the investment of researchers and R&D funds [49]. Some researchers have pointed out that the technological innovation of enterprises is easily affected by exogenous factors, and the comparability of their results is poor; thus, it is inappropriate to take the output as the explained variable [50]. Since innovation input is mainly determined by individual enterprise operators, it may reflect whether an operator exhibits agency behavior. Therefore, the proportion of enterprise R&D investment in operating revenue is used to measure the enterprise innovation capability, which is denoted as Innovation.

3.3.3. Intervening Variable

Enterprise risk-taking capacity (*Risk*). Referring to previous research literature [19,51], the asset–liability ratio is adopted to measure the risk-taking capacity of enterprises.

### 3.3.4. Control Variable

Based on the research results of Zhang et al. (2021) [47] and Zhang et al. (2015) [48], the following control variables are selected. (1) Company size (*Size*) is measured as the natural logarithm of total assets at the end of the year. (2) A company's listing age (*Age*) is measured as the natural logarithm of the interval between the company's listing date and 31 December 2020. (3) Ownership concentration (*Herf*) is measured as the sum of the squared shareholding proportions of the top five shareholders. (4) The regional economic development level (*GDP*) is measured as the natural logarithm of per capita regional GDP. (5) The openness level (*FDI*) is the natural logarithm of per capita foreign direct investment. (6) The degree of financial support (*Finance*) is measured as the proportion of added value pertaining to the financial industry in the GDP of the region. Table 1 shows the specific definitions and descriptive statistics of the variables used in this paper. See Table 2 for the correlation analysis. The results show that enterprise performance (*JROA*) has a mean value of 0.026, a maximum value of 7.446 and a minimum value of −7.700, with a standard deviation of 0.154, indicating that there are significant differences in enterprise performance between different listed enterprises. In terms of enterprise technological Innovation, the mean value is 0.043, the minimum value is 0.000, the maximum value is 2.516 and the standard deviation is 0.057, which fully indicates that there are great differences in technological innovation among different listed enterprises. The mean, minimum, maximum and standard deviation of enterprise risk-taking (*Risk*) are 0.446, 0.014, 11.386 and 0.245, respectively. It can be seen that there are great differences in enterprise risk-taking ability among different companies. In addition, from the perspective of control variables, there are also obvious differences in the age of a company listing (*Age*), ownership concentration (*Herf*), regional economic development level (*GDP*), openness level (*FDI*) and financial support degree (*Finance*).

**Table 1.** Definitions of the relevant variables.

| Variable Name | Variable Definition | Basic Statistics | | | | |
| --- | --- | --- | --- | --- | --- | --- |
| | | Sample Size | Mean Value | Standard Deviation | Minimum Value | Maximum Value |
| *JROA* | Return on total assets, net profit/total assets | 10,494 | 0.026 | 0.154 | −7.700 | 7.446 |
| *Innovation* | Enterprise technology innovation, R&D investment/operating income | 10,494 | 0.043 | 0.057 | 0.000 | 2.516 |
| *Risk* | The asset liability ratio, gross liabilities/total assets | 10,494 | 0.446 | 0.245 | 0.014 | 11.386 |
| *Size* | Enterprise scale, the natural log of total assets at year end | 10,494 | 22.440 | 1.296 | 18.393 | 28.636 |
| *Age* | Age at listing; 31 December 2020 minus the launch date | 10,494 | 2.381 | 0.599 | 0.586 | 3.403 |
| *Herf* | Ownership concentration, the sum of the squared holdings of the top five shareholders | 10,494 | 0.150 | 0.111 | 0.003 | 0.794 |
| *GDP* | Regional economic development level, the natural logarithm of GDP per capita | 10,494 | 11.133 | 0.460 | 9.873 | 12.013 |
| *FDI* | Openness, the natural logarithm of foreign direct investment per capita | 10,494 | 8.471 | 1.212 | 5.394 | 12.511 |
| *Finance* | Degree of financial support, added value of the financial industry/GDP | 10,494 | 0.084 | 0.041 | 0.027 | 0.199 |

**Table 2.** Correlation analysis.

| Variable | JROA | Innovation | Risk | Size | Age | Herf | GDP | FDI | Finance |
|---|---|---|---|---|---|---|---|---|---|
| JROA | 1.000 | | | | | | | | |
| Innovation | −0.072 *** | 1.000 | | | | | | | |
| Risk | −0.414 *** | −0.138 *** | 1.000 | | | | | | |
| Size | 0.044 *** | −0.176 *** | 0.378 *** | 1.000 | | | | | |
| Age | −0.045 *** | −0.117 *** | 0.268 *** | 0.344 *** | 1.000 | | | | |
| Herf | 0.075 *** | −0.155 *** | 0.085 *** | 0.317 *** | −0.035 *** | 1.000 | | | |
| GDP | −0.024 ** | 0.112 *** | −0.015 | 0.146 *** | 0.100 *** | −0.028 *** | 1.000 | | |
| FDI | −0.023 ** | 0.102 *** | −0.026 *** | 0.115 *** | 0.081 *** | −0.012 | 0.926 *** | 1.000 | |
| Finance | −0.026 ** | 0.104 *** | 0.012 | 0.189 *** | 0.099 *** | 0.057 *** | 0.741 *** | 0.711 *** | 1.000 |

Note: *** and ** indicate that the corresponding regression results are significant at the 1% and 5% levels, respectively.

As can be seen from the results of the correlation analysis in Table 2, the correlation coefficients among the core variables selected for empirical analysis in this paper are all less than 0.5, indicating that the multiple linear regression model constructed in this paper does not have a serious multicollinearity problem. In addition, technological innovation is significantly negatively correlated with firm performance at the significance level of 1%, indicating that the improvement of the technological innovation capability of firms may have a negative impact on firm performance, which is consistent with the conclusion of Hypothesis 1.

## 4. Positive Economics

### 4.1. Benchmark Regression Results

Model (1) in Table 3 reports the benchmark regression results regarding the impact of technological innovation on firm performance. In column 1 of model (1), the estimated coefficient of the core explanatory variable, enterprise technological innovation (*Innovation*), is significantly negative, showing that enterprise technological innovation inhibits the improvement of firm performance. This empirical result verifies hypothesis 1. In addition, in column 2, which includes the control variables, there is a significant positive correlation between enterprise size (*Size*) and firm performance, indicating that firm performance is effectively improved with an increase on the enterprise scale. The coefficient of listing age (*Age*) is significantly negative; an increase in enterprise listing age does not improve firm performance, possibly because of stereotypical thinking about technological innovation [52]. It is difficult to adjust in a timely manner according to changing market conditions. The development of innovative technology causes enterprises to fail to update their technical systems in a timely manner. Therefore, it has a negative impact on the improvement of firm performance. The coefficient of ownership concentration (*Herf*) is positive but nonsignificant, indicating that corporate performance does not improve with an increase in ownership concentration. The regional economic development level (*GDP*) and openness level (*FDI*) also fail to pass the significance test of 10%. Moreover, the negative value of the latter means that the more open a city is, the less conducive it is to the improvement of local firm performance. There is a negative correlation between the degree of financial support (*Finance*) and firm performance, but this correlation is not significant, indicating that the construction of a high-level financial capital market is not conducive to the improvement of the local firm performance.

**Table 3.** Test results regarding the mechanism of the effect of technological innovation on firm performance.

| Variable | Explained Variable JROA (1) | | Explained Variable Risk (2) | Explained Variable JROA (3) |
|---|---|---|---|---|
| Innovation | −2.272 *** (0.038) | −0.265 *** (0.038) | 0.094 ** (0.042) | −0.223 *** (0.034) |
| Risk | | | | −0.444 *** (0.008) |
| Size | | 0.027 *** (0.004) | 0.025 *** (0.005) | 0.038 *** (0.004) |
| Age | | −0.047 *** (0.011) | 0.192 *** (0.012) | 0.038 *** (0.01) |
| Herf | | 0.051 (0.036) | 0.177 *** (0.039) | 0.13 *** (0.031) |
| GDP | | 0.011 (0.025) | −0.018 (0.027) | 0.003 (0.022) |
| FDI | | −0.01 (0.006) | 0.001 (0.007) | −0.009 * (0.006) |
| Finance | | −0.089 (0.175) | 0.101 (0.192) | −0.044 (0.153) |
| Constant | 0.05 *** (0.005) | −0.496 *** (0.253) | −0.352 (0.277) | −0.653 *** (0.221) |
| Individual fixed effects | Yes | Yes | Yes | Yes |
| Year fixed effects | Yes | Yes | Yes | Yes |
| Sample size | 10,494 | 10,494 | 10,494 | 10,494 |
| $R^2$ | 0.015 | 0.022 | 0.048 | 0.254 |

Note: Robust standard errors are reported in parentheses in this table. ***, ** and * indicate that the corresponding regression results are significant at the 1%, 5% and 10% levels, respectively. The following tables are the same.

From the perspective of an enterprise's risk-taking ability, the mechanism of the effect of technological innovation on firm performance is theoretically analyzed. To verify the proposed hypothesis regarding the influence mechanism, this paper uses a mediation model for an empirical test, and the regression results are shown in Table 3. On the basis of model (1), which verifies that technological innovation has a negative impact on firm performance, model (2) determines whether technological innovation promotes a firm's risk-taking ability, and the regression coefficient of technological innovation ability is significantly positive. Finally, the mediating variable of enterprise risk-taking ability is reintroduced to the regression equation regarding the influence of enterprise technological innovation on firm performance, and the coefficient value and significance of the change in the core variable are observed. In model (3), the influence coefficient of technological innovation capability on firm performance is larger than that in model (1), indicating that risk-taking capability is the mechanism by which technological innovation hinders the improvement of firm performance. This empirical result supports hypothesis 2.

### 4.2. Robustness Test

To further test the robustness of the above empirical results, control variables are added to evaluate the robustness of the regression results. Due to differences in enterprise locations [53] and industrial development levels [54,55], the empirical results may not be robust. In this paper, the proportion of added value corresponding to the tertiary industry in GDP is selected to measure the industrial development level and to control for this factor, and the enterprise location is controlled. The results in Table 4 show that, after the industrial development level and enterprise location are controlled, the sign of the coefficient of the effect of technological innovation on firm performance remains unchanged, and this coefficient remains significant.

**Table 4.** Robustness test.

| Variable | Explained Variable JROA (6) | Explained Variable Risk (7) | Explained Variable JROA (8) |
|---|---|---|---|
| *Innovation* | −0.265 *** | 0.093 ** | −0.224 *** |
| | (0.038) | (0.042) | (0.034) |
| Control variable | Yes | Yes | Yes |
| *Constant* | −0.492 * | −0.522 * | −0.723 *** |
| | (0.278) | (0.305) | (0.243) |
| Individual fixed effects | Yes | Yes | Yes |
| Year fixed effects | Yes | Yes | Yes |
| Sample size | 10,493 | 10,494 | 10,494 |
| $R^2$ | 0.024 | 0.051 | 0.255 |

Note: ***, ** and * indicate that the corresponding regression results are significant at the 1%, 5% and 10% levels, respectively.

*4.3. Endogeneity Test*

There may be a reciprocal causal relationship between enterprise technological innovation and firm performance, which could lead to an endogeneity problem. Therefore, the lagged term of the enterprise technological innovation index is used as an instrumental variable to perform an endogeneity test and the two-stage least squares method (2SLS) is adopted for model re-estimation. First, the endogeneity test is conducted. Table 5 shows that the *p*-values of the endogeneity test reject the null hypothesis "all explanatory variables are exogenous" at the 10% level of significance, confirming the existence of endogenous explanatory variables and that the use of instrumental variables is suitable. Second, the F-statistic of the first stage is much greater than 10, reflecting the validity of the instrumental variables; that is, there are no weak instrumental variables. Finally, to enhance the robustness of the estimation results, the limited information maximum likelihood method (LIML), which is less sensitive to weak instrumental variables, is adopted. The estimation results are shown in Table 5. The estimated results of the 2SLS and LIML methods are basically consistent, the inhibitory effect of technological innovation on firm performance still holds and the estimated results are significant at the 1% level. This indicates that the empirical conclusion is unchanged after the endogeneity problems are considered.

**Table 5.** Endogeneity test.

| Variable | 2SLS (9) | LIML (10) |
|---|---|---|
| *Innovation* | −0.220 *** | −0.220 *** |
| | (0.049) | (0.049) |
| Control variable | Yes | Yes |
| *Constant* | −0.091 | −0.091 |
| | (0.090) | (0.090) |
| Individual fixed effects | Yes | Yes |
| Year fixed effects | Yes | Yes |
| Sample size | 9327 | 9327 |
| $R^2$ | 0.014 | 0.014 |
| *p* value of endogeneity test | 0.09 | 0.09 |
| F test | 5148.19 | 5148.19 |

Note: *** indicate that the regression results are significant at the 10% level.

*4.4. Further Development: Investigating the Heterogeneity of Enterprise Firm Ownership*

The dominant force of China's national economy is the state-owned economy. Through comparison, the existing literature concludes that the technological innovation efficiency of Chinese state-owned enterprises is lower than that of non-state-owned enterprises [56,57],

and Chinese enterprises are greatly influenced by the "enterprise innovation theory" determined by the nature of ownership. Previous studies believed that technological innovation could bring competitive advantages to enterprises but ignored that, when enterprises do not make enough profits in innovation activities, it is difficult to offset the huge upfront costs, which will have a negative impact on the enterprise performance [58]. In the context of China, as the firm ownership of enterprises impacts their corporate governance and corporate performance [59], the sample is divided according to the firm ownership of the sample enterprises—that is, into groups of state-owned and non-state-owned enterprises (The ownership of state-owned enterprises is determined according to the calculation results of their equity control chains; a value of "1" is assigned to state-owned enterprises, while a value of "0" is assigned to non-state-owned enterprises.). In this way, the heterogeneous impact of technological innovation on firm performance is investigated in the sample groups of state-owned enterprises and non-state-owned enterprises. As indicated by the test results in Table 6, in the sample group of non-state-owned enterprises, technological innovation significantly inhibits the improvement of the firm performance; however, this inhibitory effect is not significant in the sample group of state-owned enterprises. Foreign studies have also shown that technological innovation is negatively correlated with enterprise performance, but no in-depth analysis has been conducted from the property rights [26]. A possible reason for this result is that, compared with state-owned enterprises, non-state-owned enterprises give more attention to their technological innovation. In the current context of economic policy uncertainty, the technological innovation capacity of non-state enterprises is more fully unleashed. Although the innovation mechanism of China's private enterprises is more flexible than that of state-owned enterprises, due to their small average size and lack of strong capital chain support, the development path of technological innovation to improve enterprise performance is restricted. The results are helpful for Chinese enterprises to rationally carry out independent innovation activities according to their own conditions, so as to maintain the sustainable development of enterprises.

**Table 6.** Heterogeneity test of the impact of technological innovation on firm performance.

| Variable | Non-State-Owned Enterprises (4) | State-Owned Enterprises (5) |
|---|---|---|
| *Innovation* | −0.586 *** | −0.019 |
| | (0.077) | (0.021) |
| Control variable | Yes | Yes |
| *Constant* | −0.825 * | −0.297 * |
| | (0.462) | (0.160) |
| Individual fixed effects | Yes | Yes |
| Year fixed effects | Yes | Yes |
| Sample size | 6014 | 4480 |
| $R^2$ | 0.036 | 0.018 |

Note: *** and * indicate that the corresponding regression results are significant at the 1% and 10% levels, respectively.

## 5. Conclusions and Recommendations

Innovation is an important driving force of national economic development and an indispensable component of the creation of competitive advantages for enterprises. This study first analyzed the relationship between technological innovation and firm performance through empirical analysis and then proved that technological innovation is negatively correlated with firm performance; that is, it showed that innovation activities inhibit firm performance. In the period of transformation and upgrading, Chinese enterprises often lack key core technologies. In order to achieve technological breakthroughs, China's current technological innovation is mainly based on introduction, and the way of "market for technology" still occupies the mainstream. This indicates that, even when enterprises invest many resources into innovation activities, they are unable to realize innovation re-

sults. However, from a long-term perspective, the transformation ability of the innovation results of enterprises is low, and the performance of enterprises significantly improves. Second, this study discusses the role of the mediating transmission mechanism of corporate risk-taking in the relationship between technological innovation and firm performance, providing a new perspective for understanding how technological innovation affects firm performance and offering some ideas for subsequent research on technological innovation. Under the strategic background of China's efforts to build an innovation-oriented country, enterprises are bound to cater to the national technological innovation policy and increase the investment intensity of technological innovation. However, this may lead enterprises into the situation of excessive risk, which is not conducive to the sustainable development of enterprise performance. Finally, the influence of enterprise property heterogeneity is further investigated, and the sample is divided into state-owned enterprises and non-state-owned enterprises. The results show that, for non-state-owned enterprises, technological innovation has a significant inhibitory effect on firm performance. However, for state-owned enterprises, this inhibitory effect does not hold. There are differences in the organizational structure, hierarchy, operating model and investment financing methods between Chinese state-owned enterprises and non-state-owned enterprises. Compared to state-owned enterprises, non-state-owned enterprises will focus more on technological breakthroughs, resulting in the better release of technological innovation capabilities, which is the main reason for the less pronounced inhibiting effect of state-owned enterprises.

In view of the above research conclusions, the following suggestions are made.

First, enterprise R&D management should be standardized and strengthened. From the perspective of enterprises, in the context of innovation activities, enterprises should not only pay attention to initial investments but also evaluate subsequent R&D and output and prioritize the conversion rate of innovation achievements to drive performance growth. Only with a relatively strong ability to transform scientific research achievements can enterprises effectively improve their productivity to increase their economic benefits related to technological innovation. First, enterprises should actively absorb external resources, especially the support of government policies, to alleviate the financial pressures induced by their innovation activities and, thus, improve firm performance. Second, enterprises should improve their commercialization ability of technological innovation, closely monitor innovation output and strive to achieve cost and first-mover advantages so that they can enjoy excess profits. From the perspective of the government, it is necessary to strengthen technological innovation guidance and actively improve the innovation ability of enterprises. In addition to providing differentiated innovation subsidies to enterprises and increasing support to enterprises, the government should help enterprises establish scientific management systems for technological innovation, summarize their successful technological innovation experiences and enhance the experience of advanced enterprises through experience exchange meetings and training courses.

Second, the market and technological risks associated with technological innovation should be reduced. Since innovation activities are characterized by high risk, enterprises must conduct rigorous project demonstrations before investing, take market demand as a starting point and select highly applicable technology as innovation breakthroughs. Moreover, it is necessary to strengthen the cooperation with venture capital institutions to obtain professional guidance and thus reduce operational and investment risks. Additionally, each enterprise should measure its own comprehensive operation status and resource level, formulate a scientific and reasonable innovation input plan, reduce the risk losses, effectively improve the firm performance and optimize the capital structure. Second, the government should give attention to protecting the intellectual firm ownership of enterprises, establishing and improving laws and regulations related to intellectual firm ownership protection and protecting and maintaining the benefits of enterprise innovation to improve the firm performance. Ultimately, the government should give attention to the cultivation of innovative science and technology talent and absorption and transmission mechanisms; remain committed to promoting production–study–research cooperation;

make colleges, universities and scientific research institutions participate in enterprise technology innovation activities in the service of the real economy; safeguard enterprise innovation human capital adequacy; reduce the pressure of enterprise innovation and help enhance the level of enterprise technology innovation.

Third, the modern enterprise system must be improved. The actual income generated by the technological innovation of enterprises is not determined by the nature of their firm ownership. Enterprises with different ownership structures can prevent the principal-agent problem in enterprise innovation as much as possible by improving their corporate governance structure to improve the firm performance. The restraining effect of technological innovation on firm performance is not significant in state-owned enterprises, which may stem from the long-term accumulation and release of the reform system by state-owned enterprises; this makes state-owned enterprises eliminate ownership determinism in innovation, not only improving their innovation motivation but also effectively promoting firm performance. Therefore, we should facilitate the key role of state-owned enterprises in innovation activities and accelerate the accurate docking of the innovation and entrepreneurial chains. However, the technological innovation level of non-state-owned enterprises still has much room for improvement, and the internal systems and management systems of enterprises should be further deepened. Enterprises should be committed to narrowing the gap between their own innovation level and that of excellent enterprises and effectively achieve profit growth. Moreover, enterprises should actively improve their professional managers and technical personnel to adjust and improve their management and scientific research management, optimize their R&D management, improve the utilization rate of their R&D funding, exhibit a diversified compensation system, and enhance the good effects of innovation and the enterprise atmosphere to improve their competitiveness and performance.

**Author Contributions:** Conceptualization, H.Z. and V.A.; methodology, H.Z.; software, H.Z.; validation, V.A.; formal analysis, V.A.; investigation, H.Z.; resources, H.Z.; data curation, H.Z.; writing—original draft preparation, H.Z. and V.A.; writing—review and editing, H.Z.; visualization, H.Z.; supervision, V.A.; project administration, V.A. and funding acquisition, H.Z. All authors have read and agreed to the published version of the manuscript.

**Funding:** This research received no external funding.

**Institutional Review Board Statement:** Not applicable.

**Informed Consent Statement:** Not applicable.

**Data Availability Statement:** Publicly available datasets were analyzed in this study. This data can be found here: https://www.gtarsc.com (accessed on 25 February 2022) and http://www.stats.gov.cn/ (accessed on 25 February 2022).

**Acknowledgments:** Thanks to each of the authors for their contributions to this article.

**Conflicts of Interest:** The authors declare no conflict of interest. The funders had no role in the design of the study; in the collection, analyses or interpretation of the data; in the writing of the manuscript or in the decision to publish the results.

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
