# Peer review of "Technological Innovation, Risk-Taking and Firm Performance—Empirical Evidence from Chinese Listed Companies"

_sustainability, doi:10.3390/su142214688_

Round 1

Reviewer 1 Report

Dear authors, 

many thanks for your research, it was a pleasure for me to review it.

Please have a close look to the layout of the paper and some misspellings, like on line 264 the return on equity is used twice, there must be a mistake.

The first page with the name of the paper is empty, I believe there was a formatting mistake. 

with best regards

Author Response

Response to Reviewer 1 Comments

Respected reviewer,

Thanks very much for taking your time to review this manuscript entitled Technological Innovation, Risk Taking and Firm Performance——Empirical Evidence from Chinese Listed Companies (Manuscript ID: sustainability-1940527). I really appreciate all your comments and suggestions. I carefully revised and improved the paper as follows ( the blue font refers to the suggestions of reviewers, the black font refers to the reply instructions, and the red font refers to the detailed modifications in the paper).

Point 1: Please have a close look to the layout of the paper and some misspellings, like on line 264 the return on equity is used twice, there must be a mistake.

Response 1: Thanks for your advice, and we have re-checked the spelling of the entire paper. The misspelling on line 264 has been corrected. We also check and modify the full text of the article, Some examples are as follows:

Original text: (such as return on sales, return on equity, return on equity, or return on assets).

Revise: (such as return on sales, return on equity, return on equity, return on shareholders' equity, or return on assets).

Original text: Scientific research and experimental development (R&D), as necessary expenditures of enterprises, are mainly used to guarantee a source of funds for enterprise innovation.

Revise: Scientific research and experimental development (R&D), Research and development (R&D), as necessary expenditures of enterprises, are mainly used to guarantee a source of funds for enterprise innovation.

Original text: Due to differences in enterprise environments and capital structures, the research results of these studies do not reflect a clear conclusion.

Revise: Due to differences in enterprise environments culture and capital structures, the research results the findings of these studies do not reflect a clear conclusion.

Original text: Different subjects exhibit different fluctuations in the market environment and sometimes even have diametrically opposite effects.

Revise: Different subjects exhibit different fluctuations in the market environment. Fluctuations in market conditions can vary between firms and sometimes even have diametrically opposite effects.

Original text: However, as the main force driving policies of mass entrepreneurship and innovation, private enterprises are stronger than state-owned enterprises in terms of return on R&D investment and technological innovation; however, they also face a higher level of risk[10][11].

Revise: However, as the main force driving in responding to policies of mass entrepreneurship and innovation, private enterprises are stronger than state-owned enterprises in terms of return on R&D investment and technological innovation; however, but they also face a higher level of risk[10][11].

Original text: Table 5 shows that the p values of the endogeneity test reject the null hypothesis, namely, "all explanatory variables are exogenous", at the significance level of 10%, confirming the existence of endogenous explanatory variables and that the use of instrumental variables is suitable.

Revise: Table 5 shows that the p values of the endogeneity test reject the null hypothesis namely,"all explanatory variables are exogenous" at the significance level of 10%, 10% level of significance, confirming the existence of endogenous explanatory variables and that the use of instrumental variables is suitable.

Point 2: The first page with the name of the paper is empty, I believe there was a formatting mistake.

Response 2: Thank you very much, and we have modified the format of the first page of the paper and checked the format of the paper. The format of all tables has also been modified.

Reviewer 2 Report

Dear Authors,

The paper addresses an important and currently discussed topic of relation between technological innovation, risk taking and firm performance. However, the article needs major corrections. Below I enclose my remarks:

1.     Please, explain what is the relation with the scope of Sustainability journal.

2.     The introduction should briefly place the study in a broad context, so please present the research gap. Please highlight why this work is important and define the significance of work.

3.     Please, explain if the sample research sample is representative. I suggest to describe the research sample: size of company, branch, duration on market etc. Please explain what was the size of research sample? Why the research sample is not the same in 3.1. Sample selection and data sources and Table 1?

4.     Please add comments to Table 1 and 2, as the results should be well described.

5.     The article lacks the discussion part presenting the obtained result in comparison with the previous research conducted  in China or different countries. It would be valuable to show the specificity of Chinese companies.

6.    Conclusion section is too sketchy, the discussion should directly indicate that obtained results are related to Chinese companies.

7.     There are many typos and technical mistakes in main text, so I suggest to review the whole article. The English style is sometimes too colloquial.

Author Response

Response to Reviewer 2 Comments

Respected reviewer,

Thanks very much for taking your time to review this manuscript entitled Technological Innovation, Risk Taking and Firm Performance——Empirical Evidence from Chinese Listed Companies (Manuscript ID: sustainability-1940527). I really appreciate all your comments and suggestions. I carefully revised and improved the paper as follows ( the blue font refers to the suggestions of reviewers, the black font refers to the reply instructions, and the red font refers to the detailed modifications in the paper).

Point 1: Please, explain what is the relation with the scope of Sustainability journal.

Response 1: First of all, technology is an important factor affecting enterprises to achieve sustainable development. As a key way of enterprise transformation and upgrading, technological innovation is an effective measure for enterprises to achieve continuous improvement of performance. This paper takes technological innovation as the theme, explores whether the innovation and development of enterprises can bring competitive advantages for enterprises, help enterprises to improve the ability to prevent risks, so as to achieve sustainable development of enterprises. Secondly, the topic of our paper fits well with the theme of the special issue on sustainability, "sustainable future innovation management and organizational performance" which philosophy is that innovative practices must focus on reconciling a company's economic performance with social and ecological benefits. This paper takes listed companies as the research object, explore the enterprise technology innovation, risk-taking and the complex relation of enterprise performance. It is helpful to promote the sustainable development of enterprise strategic management. Finally, I will also improve the abstract of the paper to make it more closely related to the theme of sustainable development. Therefore, we have made the following adjustments:

Original text: This paper focuses on the practical issue of the impact of technology innovation on firm performance. Taking 1166 listed companies in China from 2012 to 2020 as research samples, this study systematically investigates and reveals the impact of technological innovation on firm performance and its internal impact mechanism. The research shows that technological innovation significantly reduces firm performance, and this conclusion holds after an endogeneity test and a robustness test. An analysis of the mechanism of action shows that risk taking is an important transmission channel through which technological innovation affects firm performance and that technological innovation can further reduce firm performance by improving risk-taking capacity. Finally, a heterogeneity test regarding the nature of firm ownership shows that technological innovation has a significantly stronger negative impact on the performance of non-state-owned enterprises than on that of state-owned enterprises. Relevant government departments and market subjects should fully understand and give attention to the impact of enterprise technological innovation on firm performance and its mechanism, which has important practical significance for standardizing and strengthening enterprise R&D management, reducing the market and technological risks of enterprise technological innovation, and perfecting modern enterprise systems.

Revise: Technological innovation can restructure the production factors of enterprises, and it is an important factor for enterprises to meet market demand, improve competitiveness, form long-term competitive advantages and obtain sustainable development. This paper focuses on the practical issue of the impact of technology innovation on firm performance. Taking 1166 listed companies in China from 2012 to 2020 as research samples, this study systematically investigates and reveals the impact of technological innovation on firm performance and its internal impact mechanism. The research shows that technological innovation significantly reduces firm performance, and this conclusion holds after an endogeneity test and a robustness test. An analysis of the mechanism of action shows that risk taking is an important transmission channel through which technological innovation affects firm performance and that technological innovation can further reduce firm performance by improving risk-taking capacity. Finally, a heterogeneity test regarding the nature of firm ownership shows that technological innovation has a significantly stronger negative impact on the performance of non-state-owned enterprises than on that of state-owned enterprises. Relevant government departments and market subjects should fully understand and give attention to the impact of enterprise technological innovation on firm performance and its mechanism, which has important practical significance for standardizing and strengthening enterprise R&D management, reducing the market and technological risks of enterprise technological innovation, and perfecting modern enterprise systems, it is helpful for firms to form a sustainable technology innovation cycle development mode.

Point 2: The introduction should briefly place the study in a broad context, so please present the research gap. Please highlight why this work is important and define the significance of work.

Response 2: In the first paragraph of the introduction, it is introduced that under the background of innovation-driven strategy and "double-cycle" development, in order to maintain sustainable competitiveness, Chinese enterprises have to embark on the road of technological innovation, promote the transformation and upgrading of enterprises through innovation, make enterprises form competitive advantages, and promote the long-term and stable development of Chinese enterprises. But technological innovation activities have the characteristics of long cycle, large investment and high risk. At present, the technological innovation capability of Chinese enterprises is insufficient, and there is still a certain gap with that of developed countries. Under this situation, it is of great significance to study the influence of technological innovation input of Chinese enterprises on enterprise performance. The research gap and significance of the paper have been added in the introduction, which is modified as follows:

Original text: In summary, the literature has made some achievements in the study of the relationships between technological innovation, firm performance and risk taking. Results based on different research perspectives, research objectives and research subjects are often not corroborated or even contradictory[12][13]. Thus, this paper further discusses the influence mechanism of technological innovation on firm performance under the difference of property right nature, this study selects state-owned enterprises and non-state-owned enterprises as research objects and conducts a dialectical analysis of the impact of technological innovation in different organizational situations to provide practical guidance for the technological innovation behaviour of state-owned and non-state-owned enterprises in uncertain environments.

Revise: In summary, the literature has made some achievements in the study of the relationships between technological innovation, firm performance and risk taking. However, due to different factors such as capital structure and industry, the conclusions drawn are not the same [7][8][9][26]. Scholars have mostly studied the relationship between risk-taking and the mediating role of risk-taking from the perspectives of executive characteristics, equity governance and executive compensation [27][28][29]. Few studies have taken risk-taking as a mediator to study the role of risk taking in the relationship between technological innovation and firm performance. Results based on different research perspectives, research objectives and research subjects are often not corroborated or even contradictory[12][13]. Thus, this study selected 1166 listed companies in China from 2012 to 2020 as research samples to explore the relationship between technological innovation, risk taking and corporate performance. Thus, based on the existing research, this paper further discusses the influence mechanism of technological innovation on firm performance under the difference of property right nature, this study selects state-owned enterprises and non-state-owned enterprises as research objects and conducts a dialectical analysis of the impact of technological innovation in different organizational situations to provide practical guidance for the technological innovation behaviour of state-owned and non-state-owned enterprises in uncertain environments. In recent years, China is in an important period of transformation and upgrading with rapid economic growth. Discussing the impact of technological innovation on Chinese enterprise performance, it is conducive to the long-term and stable improvement of Chinese enterprise performance, the maintenance of sustainable economic development of enterprises and the promotion of international competitiveness of enterprises. At the same time, the research from the new perspective of risk taking enriches the existing theories and provides a theoretical basis for enterprises in other developing countries to carry out technological innovation and change reasonably and effectively.

Point 3: Please, explain if the sample research sample is representative. I suggest to describe the research sample: size of company, branch, duration on market etc. Please explain what was the size of research sample? Why the research sample is not the same in 3.1. Sample selection and data sources and Table 1?

Response 3: Following the reviewer's suggestions, we further described the study samples, including the company scale and business scope. In addition, we have explained the size of the study sample. The reason for the inconsistency of the study sample in 3.1 is that we did not explain clearly in the paper. 1166 is the number of listed companies we finally determined, while 10494 is the total observations of these 1166 listed companies from 2012 to 2020. According to your suggestions, we have made the following modifications:

Original text: In this paper, relevant data of listed companies from 2012 to 2020 were selected and processed as follows. (1) According to the 2021 industry classification of the China Securities Regulatory Commission (CSRC), financial companies such as those providing capital market services, monetary and financial services, or insurance, as well as firms in other financial industries, were excluded. (2) Companies delisted between January 1, 2012 and December 31, 2020 were excluded, and companies listed after December 31, 2012 were excluded. (3) ST companies, *ST companies and other companies with abnormal financial conditions were excluded. Finally, (4) companies with missing data pertaining to our research variables, such as our explained variables, explanatory variables, mediating variables and control variables, for the examined company/year observations were deleted. The data used in this paper were taken from the National Tai 'an Database and the China Statistical Yearbook, and the missing data of some indicators were supplemented based on the linear interpolation method.

Revise: In this paper, relevant data of listed companies from 2012 to 2020 were selected and processed as follows. (1) According to the 2021 industry classification of the China Securities Regulatory Commission (CSRC), financial companies such as those providing capital market services, monetary and financial services, or insurance, as well as firms in other financial industries, were excluded. (2) Companies delisted between January 1, 2012 and December 31, 2020 were excluded, and companies listed after December 31, 2012 were excluded. (3) ST companies, *ST companies and other companies with abnormal financial conditions were excluded. Finally, (4) companies with missing data pertaining to our research variables, such as our explained variables, explanatory variables, mediating variables and control variables, for the examined company/year observations were deleted. After screening, 10494 observed data of 1166 listed companies are finally obtained as research samples, and stata15.1 is used for correlation analysis. Among them, the 1166 listed companies including Midea Group, Kweichow Moutai, China Telecom, China Petrochemical of listed companies and other well-known, and involves consumer goods, information technology, telecommunications services, industrial, energy, raw materials, manufacturing industries. The sample almost covers all of China's industry and is representative. The data used in this paper were taken from the CSMAR Database and the China Statistical Yearbook, and the missing data of some indicators were supplemented based on the linear interpolation method.

Point 4: Please add comments to Table 1 and 2, as the results should be well described.

Response 4: Following your suggestion, we describe the results in Tables 1 and 2 with the following modifications:

Original text: Based on the research results of Zhang et al. (2021) and Zhang et al. (2015), the following control variables are selected. 1) Company size () is measured as the natural logarithm of total assets at the end of the year. 2) A company's listing age () is measured as the natural logarithm of the interval between the company's listing date and December 31, 2020. 3) Ownership concentration () is measured as the sum of the squared shareholding proportions of the top five shareholders. 4) The regional economic development level () is measured as the natural logarithm of per capita regional GDP. 5) The openness level () is the natural logarithm of per capita foreign direct investment. Finally, 6) the degree of financial support () is measured as the proportion of added value pertaining to the financial industry in the GDP of the region. Table 1 shows the specific definitions and descriptive statistics of the variables used in this paper. See Table 2 for the correlation analysis.

Revise: Based on the research results of Zhang et al. (2021) and Zhang et al. (2015), the following control variables are selected. 1) Company size () is measured as the natural logarithm of total assets at the end of the year. 2) A company's listing age () is measured as the natural logarithm of the interval between the company's listing date and December 31, 2020. 3) Ownership concentration () is measured as the sum of the squared shareholding proportions of the top five shareholders. 4) The regional economic development level () is measured as the natural logarithm of per capita regional GDP. 5) The openness level () is the natural logarithm of per capita foreign direct investment. Finally, 6) the degree of financial support () is measured as the proportion of added value pertaining to the financial industry in the GDP of the region. Table 1 shows the specific definitions and descriptive statistics of the variables used in this paper. See Table 2 for the correlation analysis. The results show that enterprise performance (JROA) has a mean value of 0.026, a maximum value of 7.446 and a minimum value of -7.700, with a standard deviation of 0.154, indicating that there are significant differences in enterprise performance between different listed enterprises. In terms of enterprise technological Innovation, the mean value is 0.043, the minimum value is 0.000, the maximum value is 2.516, and the standard deviation is 0.057, which fully indicates that there are great differences in technological innovation among different listed enterprises. The mean, minimum, maximum and standard deviation of enterprise risk-taking (Risk) are 0.446, 0.014, 11.386 and 0.245, respectively. It can be seen that there are great differences in enterprise risk-taking ability among different companies. In addition, from the perspective of control variables, there are also obvious differences in the age of company listing (Age), ownership concentration (), regional economic development level (GDP), openness level (FDI) and financial support degree (Finance).

As can be seen from the results of correlation analysis in Table 2, the correlation coefficients among the core variables selected for empirical analysis in this paper are all less than 0.5, indicating that the multiple linear regression model constructed in this paper does not have serious multicollinearity problem. In addition, technological innovation is significantly negatively correlated with firm performance at the significance level of 1%, indicating that the improvement of technological innovation capability of firms may have a negative impact on firm performance, which is consistent with the conclusion of Hypothesis 1.

Point 5: The article lacks the discussion part presenting the obtained result in comparison with the previous research conducted in China or different countries. It would be valuable to show the specificity of Chinese companies.

Response 5: We gratefully appreciate for your valuable suggestion. China has 460,000 state-owned enterprises, and the state-owned economy is the leading force in the national economy, which is quite different from foreign countries. The dominant role of state-owned economy can ensure the dominant position of China's public economy and lead China to realize socialist modernization. However, while state-owned enterprises are dominant, there will also be some disadvantages of development. Compared with non-state-owned enterprises, state-owned enterprises have the support of the state and the market is relatively stable, which leads to the weak awareness of independent technological innovation. Therefore, the study of the heterogeneous impact of technological innovation between state-owned enterprises and non-state-owned enterprises on firm performance in the Chinese context is conducive to putting forward the operation mode that conforms to the innovation and development of Chinese enterprises from the national conditions. We have also made the following adjustments for the further discussion:

Original text: As the firm ownership of enterprises impact their corporate governance and corporate performance[54], the sample is divided according to the firm ownership of the sample enterprises, that is, into groups of state-owned and non-state-owned enterprises. In this way, the heterogeneous impact of technological innovation on firm performance is investigated in the sample groups of state-owned enterprises and non-state-owned enterprises. As indicated by the test results in Table 6, in the sample group of non-state-owned enterprises, technological innovation significantly inhibits the improvement of firm performance; however, this inhibitory effect is not significant in the sample group of state-owned enterprises. A possible reason for this result is that compared with state-owned enterprises, non-state-owned enterprises give more attention to their technological innovation. In the current context of economic policy uncertainty, the technological innovation capacity of non-state enterprises is more fully unleashed.

Revise: The dominant force of China's national economy is the state-owned economy. Through comparison, the existing literature concludes that the technological innovation efficiency of Chinese state-owned enterprises is lower than that of non-state-owned enterprises[57][58], and Chinese enterprises are greatly influenced by the "enterprise innovation capability theory" determined by the nature of ownership. Previous studies believed that technological innovation can bring competitive advantages to enterprises, but ignore that when enterprises do not make enough profits in innovation activities, it is difficult to offset the huge upfront costs, which will have a negative impact on enterprise performance[59]. In the context of China, as the firm ownership of enterprises impact their corporate governance and corporate performance[60], the sample is divided according to the firm ownership of the sample enterprises, that is, into groups of state-owned and non-state-owned enterprises. In this way, the heterogeneous impact of technological innovation on firm performance is investigated in the sample groups of state-owned enterprises and non-state-owned enterprises. As indicated by the test results in Table 6, in the sample group of non-state-owned enterprises, technological innovation significantly inhibits the improvement of firm performance; however, this inhibitory effect is not significant in the sample group of state-owned enterprises. Foreign studies have also shown that technological innovation is negatively correlated with enterprise performance, but no in-depth analysis has been conducted from the property rights[26]. A possible reason for this result is that compared with state-owned enterprises, non-state-owned enterprises give more attention to their technological innovation. In the current context of economic policy uncertainty, the technological innovation capacity of non-state enterprises is more fully unleashed. Although the innovation mechanism of China's private enterprises is more flexible than that of state-owned enterprises, due to their small average size and lack of strong capital chain support, the development path of technological innovation to improve enterprise performance is restricted. The results are helpful for Chinese enterprises to rationally carry out independent innovation activities according to their own conditions, so as to maintain the sustainable development of enterprises.

Point 6: Conclusion section is too sketchy, the discussion should directly indicate that obtained results are related to Chinese companies.

Response 6: We thank the reviewer for pointing this out. We have revised.

Original text: Innovation is an important driving force of national economic development and an indispensable component of the creation of competitive advantages for enterprises. This paper first analyses the relationship between technological innovation and firm performance through empirical analysis and then proves that technological innovation is negatively correlated with firm performance; that is, it shows that innovation activities inhibit firm performance. This indicates that even when enterprises invest many resources in innovation activities, they are unable to realize innovation results. However, from a long-term perspective, the transformation ability of the innovation results of enterprises is low, and the performance of enterprises significantly improves. Second, this paper discusses the role of the mediating transmission mechanism of corporate risk taking in the relationship between technological innovation and firm performance, providing a new perspective for understanding how technological innovation affects firm performance and offering some ideas for subsequent research on technological innovation. Finally, the influence of enterprise property heterogeneity is further investigated, and the sample is divided into state-owned enterprises and non-state-owned enterprises. The results show that for non-state-owned enterprises, technological innovation has a significant inhibitory effect on firm performance. However, for state-owned enterprises, this inhibitory effect does not hold.

Revise: Innovation is an important driving force of national economic development and an indispensable component of the creation of competitive advantages for enterprises. This paper first analyses the relationship between technological innovation and firm performance through empirical analysis and then proves that technological innovation is negatively correlated with firm performance; that is, it shows that innovation activities inhibit firm performance. In the period of transformation and upgrading, Chinese enterprises often lack key core technologies. In order to achieve technological breakthroughs, China's current technological innovation is mainly based on introduction, and the way of "market for technology" still occupies the mainstream. This indicates that even when enterprises invest many resources in innovation activities, they are unable to realize innovation results. However, from a long-term perspective, the transformation ability of the innovation results of enterprises is low, and the performance of enterprises significantly improves. Second, this paper discusses the role of the mediating transmission mechanism of corporate risk taking in the relationship between technological innovation and firm performance, providing a new perspective for understanding how technological innovation affects firm performance and offering some ideas for subsequent research on technological innovation. Under the strategic background of China's efforts to build an innovation-oriented country, enterprises are bound to cater to the national technological innovation policy and increase the investment intensity of technological innovation. However, this may lead enterprises into the situation of excessive risk, which is not conducive to the sustainable development of enterprise performance. Finally, the influence of enterprise property heterogeneity is further investigated, and the sample is divided into state-owned enterprises and non-state-owned enterprises. The results show that for non-state-owned enterprises, technological innovation has a significant inhibitory effect on firm performance. However, for state-owned enterprises, this inhibitory effect does not hold. There are differences in the organizational structure, hierarchy, operating model and investment financing methods between Chinese state-owned enterprises and non-state-owned enterprises. Compared to state-owned enterprises, non-state-owned enterprises will focus more on technological breakthroughs, resulting in better release of technological innovation capabilities, which is the main reason for the less pronounced inhibiting effect of state-owned enterprises.

Point 7: There are many typos and technical mistakes in main text, so I suggest to review the whole article. The English style is sometimes too colloquial.

Response 7: Thank you for your advice. We have rechecked the spelling, formatting and English expression, and we have made the following modifications:

Original text: (such as return on sales, return on equity, return on equity, or return on assets).

Revise: (such as return on sales, return on equity, return on equity, return on shareholders' equity, or return on assets).

Original text: Scientific research and experimental development (R&D), as necessary expenditures of enterprises, are mainly used to guarantee a source of funds for enterprise innovation.

Revise: Scientific research and experimental development (R&D), Research and development (R&D), as necessary expenditures of enterprises, are mainly used to guarantee a source of funds for enterprise innovation.

Original text: Due to differences in enterprise environments and capital structures, the research results of these studies do not reflect a clear conclusion.

Revise: Due to differences in enterprise environments culture and capital structures, the research results the findings of these studies do not reflect a clear conclusion.

Original text: Different subjects exhibit different fluctuations in the market environment and sometimes even have diametrically opposite effects.

Revise: Different subjects exhibit different fluctuations in the market environment. Fluctuations in market conditions can vary between firms and sometimes even have diametrically opposite effects.

Original text: However, as the main force driving policies of mass entrepreneurship and innovation, private enterprises are stronger than state-owned enterprises in terms of return on R&D investment and technological innovation; however, they also face a higher level of risk[10][11].

Revise: However, as the main force driving in responding to policies of mass entrepreneurship and innovation, private enterprises are stronger than state-owned enterprises in terms of return on R&D investment and technological innovation; however, but they also face a higher level of risk[10][11].

Original text: Table 5 shows that the p values of the endogeneity test reject the null hypothesis, namely, "all explanatory variables are exogenous", at the significance level of 10%, confirming the existence of endogenous explanatory variables and that the use of instrumental variables is suitable.

Revise: Table 5 shows that the p values of the endogeneity test reject the null hypothesis namely,"all explanatory variables are exogenous" at the significance level of 10%, 10% level of significance, confirming the existence of endogenous explanatory variables and that the use of instrumental variables is suitable.

Round 2

Reviewer 2 Report

Dear Authors,

thank you for revision of paper and corrections according to my remarks.